# Is tamoxifen good enough for the Asian population in ER+ HER2- post-menopausal women with early breast cancer?
# A nationwide population-based cohort study

**Chuan-Hsun Chang[1☺], Yi-Chan Lee[2☺], Chun-Wen Huang[3], You-Min Lu[4]\***

**1** Department of Surgery, Cheng Hsin General Hospital, Taipei, Taiwan, R.O.C, **2** Institute of Health Policy and Management, College of Public Health, National Taiwan University, Taipei, Taiwan, R.O.C, **3** Office of the Superintendent, Cheng Hsin General Hospital, Taipei, Taiwan, R.O.C, **4** Division of Clinical Pharmacy, Department of Pharmacy, Cheng Hsin General Hospital, Taipei, Taiwan, R.O.C

☺ These authors contributed equally to this work.
\* ch1306@chgh.org.tw

**Data Availability Statement:** The data that support the findings of this study are available from Taiwan's Ministry of Health and Welfare's Health and Welfare Data Center (HWDC). However, due to

## Abstract

### Background

Numerous clinical trials have compared the efficacy of endocrine therapy in post-menopausal breast cancer patients. This study aims to explore whether there is a difference in recurrence rates between this population using tamoxifen and aromatase inhibitors (AIs) by analyzing real-world data.

### Methods

This retrospective cohort study utilized the National Health Insurance (NHI) claims data and the Taiwan Cancer Registry (TCR). We identified 6,050 patients aged over 55 diagnosed with ER-positive, HER2-negative early breast cancer between 2012 and 2016 (4,451 on AIs alone and 1,599 on tamoxifen alone). Recurrence in both groups was assessed until the end of 2020. Hazards were measured based on age of diagnosis, cancer stage, adjuvant chemotherapy, radiation therapy, type of endocrine therapy used, and adherence. Recurrence-free survival between the AIs and tamoxifen groups was evaluated using the Kaplan-Meier model.

### Results

The average age was 65.1 years, with a median follow-up time of 5.7 years and a median duration of endocrine therapy of 4.5 years. The recurrence rate was 2.2%. Using tamoxifen as endocrine therapy reduces the risk of recurrence compared to AIs (adjusted HR: 0.32, p < 0.0001). There was no statistical difference between the two drugs in stage 1 breast cancer. However, in stage 2, the risk of breast cancer recurrence decreased to 0.15 times with the use of tamoxifen compared to AIs (p = 0.0002). Stage 2 cancer, histological grade 3, and non-adherence increased recurrence risk in post-menopausal breast cancer patients.

licensing restrictions, these data are not publicly accessible. Data access is restricted to individuals who register with the HWDC (https://www.apre.mohw.gov.tw/) and are Taiwanese nationals. For inquiries, please contact Mr. Yang at +886-2-85906805 or via email at stsung@mohw.gov.tw. Detailed descriptions of the databases used are provided in the Methods section, and the corresponding codebook is accessible at https://dep.mohw.gov.tw/DOS/lp-2503-113.html. By following the procedures outlined in the Methods section, the findings of this study can be replicated.

**Funding:** YES. This study was funded by the Cheng Hsin General Hospital (CHGH110-(N)21). The funders had no role in study design, data collection and analysis, decision to publish, or preparation of the manuscript.

**Competing interests:** The authors have declared that no competing interests exist.

## Conclusion

Based on real-world data analysis, in ER-positive, HER2-negative post-menopausal women with early breast cancer in Taiwan, the use of tamoxifen compared to AIs is associated with a lower risk of recurrence. Improved adherence to medication can break the cycle of recurrence and improve health outcomes.

## Introduction

Breast cancer treatment includes surgery, chemotherapy, endocrine therapy, radiation therapy, and targeted therapy. Endocrine therapy is recognized as an effective adjuvant treatment for patients with hormone receptor-positive breast cancer. Endocrine therapy includes two types: tamoxifen and AIs. Tamoxifen can be used in pre-menopausal and post-menopausal patients; however, AIs are only suitable for post-menopausal breast cancer patients. The AIs available include exemestane, anastrozole, and letrozole.

Numerous clinical trials have compared the efficacy of tamoxifen and AIs in post-menopausal breast cancer patients. The ATAC trial [1], which compared the use of 5 years of anastrozole or tamoxifen, found that the anastrozole group had better disease-free survival (HR 0.91; p = 0.04), but there was no difference in overall mortality (HR 0.95; p = 0.4). Another large clinical trial, BIG 1–98 [2], compared the 5- year use of letrozole or tamoxifen and followed patients for a median of 8.1 years. The results showed that the letrozole group had better disease-free survival (HR 0.86, p = 0.007) and overall survival (HR 0.87, p = 0.048) than the tamoxifen group.

Another type of randomized controlled trial was conducted sequentially. Studies such as IES [3], ABCSG 8/ARNO [4], ITA [5], and ARNO 95 [6] administered tamoxifen for 2–3 years followed by AIs. Compared to five years of tamoxifen, these studies showed better disease-free survival. However, in terms of overall survival, only the ARNO 95 study [6] showed a significant improvement (HR 0.53, p = 0.045), while the IES study [3] did not demonstrate a significant difference.

According to the 2019 ESMO Clinical Practice Guidelines [7], AIs and tamoxifen are standard treatments for post-menopausal women. AIs can be used upfront, after 2–3 years of tamoxifen, or as extended adjuvant therapy after 5 years of tamoxifen. The 2023 NICE guideline [8] and the 2019 British Menopause Society consensus statement [9] recommend using AIs as initial treatment for post-menopausal women at medium or high risk of breast cancer recurrence. For women at low risk or those who have contraindications or intolerances to AIs, tamoxifen can be used.

Randomized controlled trials (RCTs) can be used to evaluate the efficacy of drug treatments, meaning the expected effects that drugs can achieve in specific populations meeting their inclusion criteria. However, RCT results may have limited external validity, and their application in clinical practice may be restricted due to factors such as short recruitment periods, small sample sizes, homogeneity of trial participants, and idealized drug usage conditions. In the real-world setting, patients often have different characteristics from those included in clinical trials, such as having multiple comorbidities, requiring multiple medications, having impaired liver or kidney function, or being older. Additionally, in the real world, patients may not adhere to follow-up appointments, and there is no confirmation of medication adherence. Therefore, this study uses real-world data analysis to explore whether there are differences in effectiveness between tamoxifen and AIs in post-menopausal breast cancer patients.

## Materials and methods

### Study design and data sources

This retrospective cohort study utilizes two databases: the National Health Insurance (NHI) claims data and the Taiwan Cancer Registry (TCR). The NHI program in Taiwan was launched in 1995 and covers healthcare data for over 99% of the country's 23 million population. Taiwan's National Health Insurance Research Database (NHIRD) is a population-level data source that requires investigators to conduct on-site analysis at a Health and Welfare Data Center (HWDC) established by Taiwan's Ministry of Health and Welfare (MOHW). Using encrypted patient IDs, we linked the patients with early stages breast cancer from TCR to the NHI database to obtain their medication records, including outpatient, inpatient, and contracted pharmacy data. The database was de-identified, and we could not identify individual participants during or after data collection. The TCR is a nationwide population-based cancer registry system established by the Ministry of Health and Welfare in 1979. The TCR has maintained a long-form database since 2002 to evaluate cancer care patterns and treatment outcomes, recording cancer staging, detailed treatment information, and recurrence data [10]. This study received review and approval from the Institutional Review Board of Cheng Hsin General Hospital (IRB No. CHGH-IRB 113E-03-2).

### Study population

Using the TCR, we identified patients diagnosed with breast cancer for the first time between January 1, 2012, to December 31, 2016, who had cancer stages 1–2. The study period commenced from the initial breast cancer diagnosis date until recurrence, death, or December 31, 2020. We considered the International Classification of Diseases for Oncology, Third Edition (ICD-O-3) to determine breast cancer's primary site (C50.0-C50.9). Patients with any cancer diagnosis within two years before the confirmation of breast cancer were excluded from the study.

We identified 24,916 patients who had received endocrine therapy for at least one year, and only those who exclusively used tamoxifen or AIs were included in the research. We then included 17,075 individuals who were diagnosed with cancer at stages 1–2. We used an age greater than 55 years as a surrogate indicator for menopause. Therefore, patients diagnosed with cancer at 55 years or younger (9,347 individuals) were excluded from the study. Individuals with breast tumors classified as estrogen receptor-positive (ER+) and human epidermal growth factor receptor 2- negative (HER2-) and those not requiring targeted therapy were less likely to have critically distorted results, excluding 1,631 and 47 individuals, respectively. As a result, 6,050 individuals were considered as the study subjects.

Tamoxifen can be used in pre-and post-menopausal women, while AIs are indicated explicitly for post-menopausal women. The study focused on post-menopausal breast cancer patients to ensure comparability between the two drugs. Unfortunately, the NHI database lacks information on whether patients have stopped menstruating. A meta-analysis [11] that analyzed menarche, menopause, and breast cancer risk included 117 epidemiological studies. The mean age at natural menopause was 49.3 years, and 10% of women (16,144 out of 170,413) reported menopause occurring at age 55 or older. Since 90% of patients undergo menopause before the age of 55, this study uses an age greater than 55 as a proxy indicator for post-menopausal status.

### Variables and outcome

Data were collected on the study subjects, including age at initial diagnosis of breast cancer, cancer stage, adjuvant chemotherapy, radiation therapy, histological grade, group classification

based on endocrine therapy, duration of follow-up, total duration of medication use, recurrence status, cause of death, interruptions, and adherence. Tumor staging was determined according to the seventh edition of the AJCC Cancer Staging Manual. The stage of breast cancer includes both the pathological and clinical stages. In this study, priority was given to the pathological stage. The clinical stage was used if the pathological stage was indicated as "unclear or not recorded" or unavailable.

The groups for endocrine therapy were classified based on pharmacological classes. Group I included patients who received AIs alone, which included the components anastrozole (ATC code L02BG03), exemestane (ATC code L02BG06), and letrozole (ATC code L02BG04). Group II consisted of patients who received tamoxifen alone (ATC code L02BA01). The prescription history of patients receiving endocrine therapy was examined to determine if interruptions and non-adherence occurred. Any gap period between two consecutive endocrine therapy prescriptions that exceeded 180 days was defined as an "interruption" [12, 13]. The proportion of days covered (PDC) was used as an indicator to assess patient adherence to treatment [14]. PDC was defined as the number of days covered by all medications minus the number of overlapping days, divided by the total number of days between the beginning and end of treatment [15]. Patients with a PDC greater than 80% were considered adherent. In comparison, those with less than 80% were considered non-adherent.

## Analysis

Statistical analysis was performed using the SAS 9.4 software package for each cancer stage. Descriptive statistics were used to present the distribution of continuous variables using means. In contrast, percentages were used to illustrate the differences in the distribution of categorical variables between AIs and tamoxifen, with chi-squared tests conducted for comparison.

Inferential statistics were conducted using the Cox proportional hazards model to estimate the crude hazard ratio and 95% confidence interval (CI) for recurrence in breast cancer patients. Subsequently, a multivariable Cox proportional hazards model was employed to incorporate control variables into the analysis, and the adjusted hazard ratio and 95% CI for recurrence associated with AIs and tamoxifen were observed. The significance level for statistical testing was set at $p < 0.05$.

Breast cancer recurrence-free survival (RFS) curves were generated using the Kaplan-Meier method to assess cumulative RFS probabilities. The log-rank test was then used to compare the RFS curves and determine whether they had statistically significant differences. Kaplan-Meier curves were plotted to depict the cumulative RFS probabilities of AIs and tamoxifen upon an 8-year follow-up period. Furthermore, the RFS outcomes of AIs and tamoxifen were investigated for each cancer stage. Finally, the log-rank test was employed to compare cumulative RFS probabilities, and if $p < 0.05$, it indicated a statistically significant difference.

## Results

### Characteristics of the study cohort

The study population comprised 6,050 individuals with breast cancer incidence for the first time. Among these individuals, 4,451 received treatment with AIs, while 1,599 received tamoxifen, with a 7.0% and 9.7% mortality rate, respectively. The average age of the study cohort was 65.1 ±7.8 years. Post-menopausal breast cancer stage 1–2 cases have increased from 666 in 2012 to 1,736 in 2016. Among patients receiving endocrine therapy, the use of tamoxifen decreased from 461 patients (69.2% of the total) in 2012 to only 253 patients (14.6%) in 2016.

In contrast, the proportion of patients using AIs increased annually (Cochran-Armitage trend test, Z = 26.1, p < 0.0001), as shown in Fig 1.

Analysis of patients' breast cancer stages revealed that among those receiving AIs treatment, the most common stage was stage 2 (50.6%); for patients treated with tamoxifen, stage 1 was predominant (65.8%). The median follow-up time is 5.6 years. Most patients (71.6%) were followed up for 4–7 years; 17.1% in the AIs group and 45.3% in the tamoxifen group were followed up for 7–10 years. In this study, 38.7% of patients utilized adjuvant chemotherapy and 48.9% underwent radiation therapy. The prescription duration of endocrine therapy is 3.4 years. Most patients (64.2%) received treatment for 3–5 years, while 21.6% in the AI group and 28.8% in the tamoxifen group received treatment for over five years. Patients receiving tamoxifen had higher rates of treatment interruption (11.6% vs. 8.7%) and non-adherence (14.3% vs. 9.3%) than those receiving AIs. The demographic data of the study cohort are summarized in Table 1.

## Survival and recurrence-free survival

By following up until December 31, 2020, the overall survival rate of 6,050 breast cancer patients was 92.3%. Among the 468 deceased patients, approximately 27.4% of deaths were attributed to breast cancer. Analysis of patient death time revealed that the highest number of deaths occurred within 2–5 years after breast cancer diagnosis, accounting for 57.7% (270 individuals). The second-highest number of deaths was observed after 5 years, with 148 individuals (31.6%).

This study included post-menopausal breast cancer patients, among whom the recurrence rate was 2.2% (132 individuals). When stratified by stage, the recurrence rates were 1.3% in stage 1 and 3.3% in stage 2. Regarding different endocrine therapies, the recurrence rate was 2.7% in the AIs group and 0.9% in the tamoxifen group. The highest number of recurrences occurred two years or more after the diagnosis of breast cancer, accounting for 1.3% (81 individuals). The comparison of recurrence rates between the AIs and tamoxifen groups is presented in Table 2.

For patients with stage 1 breast cancer after menopause, there was no significant difference in the recurrence-free survival rates between those treated with AIs or tamoxifen (98.6% vs.

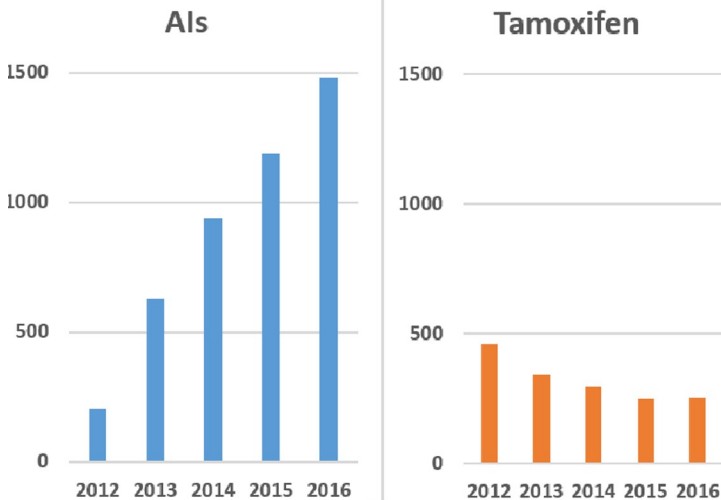

**Fig 1. Trends in endocrine therapy use among post-menopausal breast cancer patients from 2012 to 2016.**

**Table 1. Demographic data of the study cohort.**

| | Total (N = 6,050) | | AIs alone (n = 4,451; 73.6%) | | Tamoxifen alone (n = 1,599; 26.4%) | | P-values |
|---|---|---|---|---|---|---|---|
| First diagnosis year | | | | | | | < .0001 |
| 2012 | 666 | 11.0 | 205 | 4.6 | 461 | 28.8 | |
| 2013 | 971 | 16.0 | 629 | 14.1 | 342 | 21.4 | |
| 2014 | 1238 | 20.5 | 943 | 21.2 | 295 | 18.4 | |
| 2015 | 1439 | 23.8 | 1191 | 26.8 | 248 | 15.5 | |
| 2016 | 1736 | 28.7 | 1483 | 33.3 | 253 | 15.8 | |
| Breast cancer stage | | | | | | | < .0001 |
| Stage 1 | 3252 | 53.8 | 2200 | 49.4 | 1052 | 65.8 | |
| Stage 2 | 2798 | 46.2 | 2251 | 50.6 | 547 | 34.2 | |
| Histological grade | | | | | | | < .0001 |
| Grade 1 | 1661 | 27.5 | 1163 | 26.1 | 498 | 31.1 | |
| Grade 2 | 3462 | 57.2 | 2588 | 58.1 | 874 | 54.7 | |
| Grade 3 | 722 | 11.9 | 571 | 12.8 | 151 | 9.4 | |
| Unknown | 205 | 3.4 | 129 | 2.9 | 76 | 4.8 | |
| Follow-up years (until the end of follow-up or death) | | | | | | | < .0001 |
| <4 years | 232 | 3.8 | 165 | 3.7 | 67 | 4.2 | |
| 4–7 years | 4330 | 71.6 | 3523 | 79.2 | 807 | 50.5 | |
| 7–10 years | 1488 | 24.6 | 763 | 17.1 | 725 | 45.3 | |
| The primary cause of death | | | | | | | < .0001 |
| No death | 5582 | 92.3 | 4138 | 93.0 | 1444 | 90.3 | |
| Breast cancer | 128 | 2.1 | 101 | 2.3 | 27 | 1.7 | |
| Non-breast cancer | 340 | 5.6 | 212 | 4.8 | 128 | 8.0 | |
| Years of death after diagnosis | | | | | | | < .0001 |
| No deaths | 5582 | 92.3 | 4138 | 93.0 | 1444 | 90.3 | |
| <2 years | 50 | 0.8 | 33 | 0.7 | 17 | 1.1 | |
| 2–5 years | 270 | 4.5 | 197 | 4.4 | 73 | 4.6 | |
| > = 5 years | 148 | 2.4 | 83 | 1.9 | 65 | 4.1 | |
| Adjuvant chemotherapy | | | | | | | < .0001 |
| No | 3709 | 61.3 | 2549 | 57.3 | 1160 | 72.5 | |
| Yes | 2341 | 38.7 | 1902 | 42.7 | 439 | 27.5 | |
| Radiation therapy | | | | | | | < .0001 |
| No | 3092 | 51.1 | 2186 | 49.1 | 906 | 56.7 | |
| Yes | 2958 | 48.9 | 2265 | 50.9 | 693 | 43.3 | |
| Prescription duration of endocrine therapy | | | | | | | < .0001 |
| <3 years | 741 | 12.2 | 493 | 11.1 | 248 | 15.5 | |
| 3–5 years | 3886 | 64.2 | 2996 | 67.3 | 890 | 55.7 | |
| > = 5 years | 1423 | 23.5 | 962 | 21.6 | 461 | 28.8 | |
| Persistence or interruption* | | | | | | | 0.0007 |
| Persistence | 5479 | 90.6 | 4065 | 91.3 | 1414 | 88.4 | |
| Interruption* | 571 | 9.4 | 386 | 8.7 | 185 | 11.6 | |
| Interruption* (gap number) | | | | | | | < .0001 |
| no gap | 5479 | 90.6 | 4065 | 91.3 | 1414 | 88.4 | |
| gap = 1 | 435 | 7.2 | 312 | 7.0 | 123 | 7.7 | |
| gap = 2 | 82 | 1.4 | 42 | 0.9 | 40 | 2.5 | |
| gap = 3 | 32 | 0.5 | 19 | 0.4 | 13 | 0.8 | |
| gap>3 | 22 | 0.4 | 13 | 0.3 | 9 | 0.6 | |

*(Continued)*

**Table 1.** (Continued)

| | Total (N = 6,050) | | AIs alone (n = 4,451; 73.6%) | | Tamoxifen alone (n = 1,599; 26.4%) | | P-values |
|---|---|---|---|---|---|---|---|
| Time to the first gap of 180 days | | | | | | | **0.0047** |
| no gap | 5479 | 90.6 | 4065 | 91.3 | 1414 | 88.4 | |
| <1 year | 107 | 1.8 | 81 | 1.8 | 26 | 1.6 | |
| 1–2 year | 122 | 2.0 | 78 | 1.8 | 44 | 2.8 | |
| 2–3 year | 84 | 1.4 | 55 | 1.2 | 29 | 1.8 | |
| 3–4 year | 107 | 1.8 | 74 | 1.7 | 33 | 2.1 | |
| > = 4 years | 151 | 2.5 | 98 | 2.2 | 53 | 3.3 | |
| Adherence (PDC**) | | | | | | | < .0001 |
| <0.8 | 644 | 10.6 | 415 | 9.3 | 229 | 14.3 | |
| 0.8–0.9 | 1177 | 19.5 | 834 | 18.7 | 343 | 21.5 | |
| > = 0.9 | 4229 | 69.9 | 3202 | 71.9 | 1027 | 64.2 | |

\* Between two consecutive endocrine therapy prescriptions for more than 180 days

\*\* The proportion of days covered

99.0%; Log-rank test, p = 0.2313) (Fig 2). However, for patients with stage 2 breast cancer, the recurrence-free survival rate was higher in the tamoxifen group compared to the AIs group, with a statistically significant difference (96.1% vs. 99.3%; Log-rank test, p = 0.0002) (Fig 3).

## Recurrence rate and associated factors

In the multivariate analyses, the age of diagnosis increases the risk of breast cancer recurrence by 1.03 times. In different stages of breast cancer, stage 2 increases the risk of recurrence by 2.07 times compared to stage 1 (p = 0.0004). For patients with histological grade 3, the risk of breast cancer recurrence increases by 2.24 times (p = 0.0076). Analyzing the impact of different endocrine therapy, using tamoxifen reduces the risk of recurrence compared to AIs (total adjusted HR: 0.32, p < 0.0001; stage 2 adjusted HR: 0.15, p = 0.0002). Adjuvant chemotherapy

**Table 2.** Recurrence between the AIs and tamoxifen groups.

| | Total (N = 6,050) | | AIs alone (n = 4,451; 73.6%) | | Tamoxifen alone (n = 1,599; 26.4%) | |
|---|---|---|---|---|---|---|
| | N | % | N | % | N | % |
| Recurrence | | | | | | |
| No | 5918 | 97.8 | 4333 | 97.3 | 1585 | 99.1 |
| Yes | 132 | 2.2 | 118 | 2.7 | 14 | 0.9 |
| Breast cancer stage 1 | | | | | | |
| Non-recurrence | 3211 | 98.7 | 2169 | 98.6 | 1042 | 99.0 |
| Recurrence | 41 | 1.3 | 31 | 1.4 | 10 | 1.0 |
| Breast cancer stage 2 | | | | | | |
| Non-recurrence | 2707 | 96.7 | 2164 | 96.1 | 543 | 99.3 |
| Recurrence | 91 | 3.3 | 87 | 3.9 | 4 | 0.7 |
| Years of first recurrence after diagnosis | | | | | | |
| Non-recurrence | 5918 | 97.8 | 4333 | 97.3 | 1585 | 99.1 |
| <1 year | 26 | 0.4 | 23 | 0.5 | 3 | 0.2 |
| 1–2 year | 25 | 0.4 | 22 | 0.5 | 3 | 0.2 |
| > = 2 year | 81 | 1.3 | 73 | 1.6 | 8 | 0.5 |

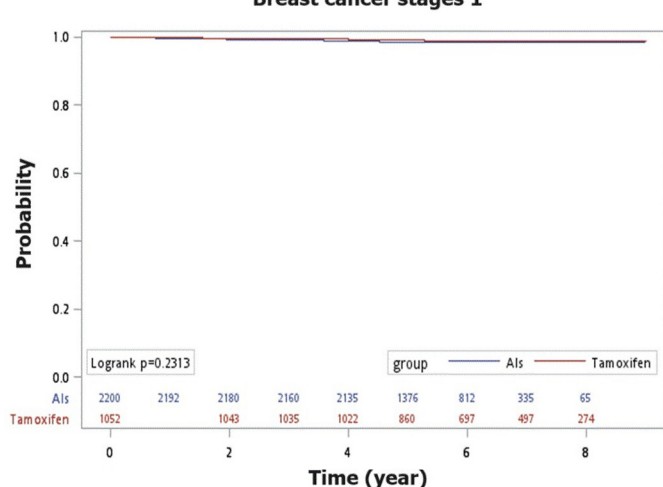

**Fig 2. The Kaplan-Meier curves depict recurrence-free survival in comparing AIs and tamoxifen for breast cancer stage 1.**

shows no statistically significant difference in recurrence. Adherence (PDC > = 0.8) decreases the risk of recurrence (adjusted HR: 0.29, p < 0.0001). Across different stages, recurrence risk rises by 0.50 times in stage 1 and 0.24 times in stage 2. The univariate and multivariable adjusted hazard ratios of covariates for recurrence are summarized in Table 3.

## Discussion

In Taiwan, the incidence of breast cancer among post-menopausal women has been steadily increasing. This study utilizes real-world data to analyze the effectiveness of tamoxifen and AIs for treating post-menopausal breast cancer. The findings suggest comparable recurrence with both endocrine therapies in stage 1 breast cancer, while tamoxifen shows superiority over AIs in stage 2.

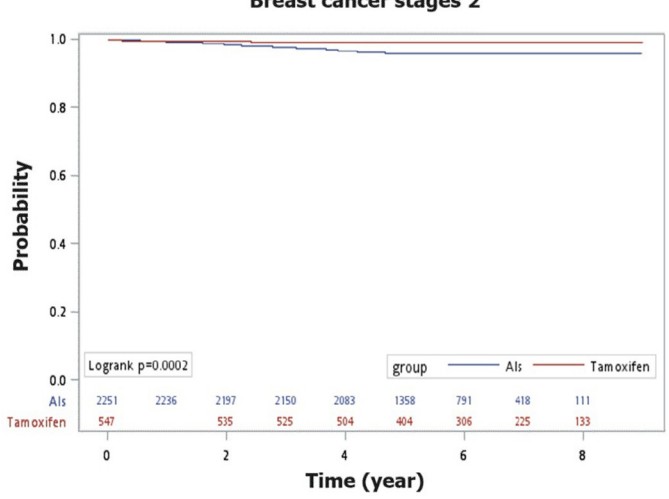

**Fig 3. The Kaplan-Meier curves depict recurrence-free survival in comparing AIs and tamoxifen for breast cancer stage 2.**

**Table 3. Univariate and multivariable adjusted hazard ratios of covariates for recurrence (N = 6,050).**

| Characteristic | Univariate HR | P-values | Adjusted HR | P-values |
|---|---|---|---|---|
| **Age of diagnosis** | 1.03(1.01–1.05) | **0.0016** | 1.03(1.00–1.05) | **0.0307** |
| **Breast cancer stage** | | | | |
| Stage 1 | 1 | | 1 | |
| Stage 2 | 2.66(1.84–3.84) | < .0001 | 2.07(1.38–3.10) | **0.0004** |
| **Histological grade** | | | | |
| Grade 1 | 1 | | 1 | |
| Grade 2 | 1.78(1.11–2.85) | **0.0165** | 1.50(0.93–2.41) | 0.0985 |
| Grade 3 | 2.76(1.57–4.88) | **0.0005** | 2.24(1.24–4.06) | **0.0076** |
| Unknown | 1.12(0.34–3.74) | 0.8540 | 0.93(0.28–3.13) | 0.9078 |
| **Endocrine therapy** | | | | |
| AIs alone | 1 | | 1 | |
| Tamoxifen alone | 0.32(0.18–0.56) | < .0001 | 0.32(0.18–0.57) | < .0001 |
| **Adherence (PDC* > = 0.8)** | | | | |
| Adherence | 0.30(0.21–0.44) | < .0001 | 0.29(0.20–0.42) | < .0001 |
| Non-adherence | 1 | | 1 | |
| **Adjuvant chemotherapy** | | | | |
| No | 1 | | 1 | |
| Yes | 1.19(0.84–1.67) | 0.3337 | 0.88(0.58–1.33) | 0.5314 |
| **Radiation therapy** | | | | |
| No | 1 | | 1 | |
| Yes | 0.74(0.52–1.04) | 0.0817 | 0.86(0.60–1.23) | 0.4120 |
| **Stage 1** | | | | |
| **Age of diagnosis** | 1.03(0.99–1.07) | 0.1266 | 1.04(1.00–1.09) | 0.0652 |
| **Histological grade** | | | | |
| Grade 1 | 1 | | 1 | |
| Grade 2 | 1.99(0.90–4.40) | 0.0916 | 1.95(0.88–4.35) | 0.1020 |
| Grade 3 | 3.91(1.42–10.78) | **0.0084** | 4.20(1.45–12.16) | **0.0081** |
| Unknown | 1.24(0.16–9.92) | 0.8392 | 1.38(0.17–11.11) | 0.7602 |
| **Endocrine therapy** | | | | |
| AIs alone | 1 | | 1 | |
| Tamoxifen alone | 0.65(0.32–1.33) | 0.2352 | 0.64(0.31–1.31) | 0.2227 |
| **Adherence (PDC* > = 0.8)** | | | | |
| Adherence | 0.55(0.24–1.23) | 0.1432 | 0.50(0.22–1.14) | 0.0993 |
| Non-adherence | 1 | | 1 | |
| **Adjuvant chemotherapy** | | | | |
| No | 1 | | 1 | |
| Yes | 1.19(0.60–2.38) | 0.6192 | 1.04(0.49–2.22) | 0.9107 |
| **Radiation therapy** | | | | |
| No | 1 | | 1 | |
| Yes | 1.14(0.62–2.12) | 0.6689 | 1.31(0.69–2.49) | 0.4144 |
| **Stage 2** | | | | |
| **Age of diagnosis** | 1.02(1.00–1.05) | 0.0525 | 1.02(0.99–1.05) | 0.1585 |
| **Histological grade** | | | | |
| Grade 1 | 1 | | 1 | |
| Grade 2 | 1.28(0.72–2.31) | 0.4037 | 1.24(0.69–2.24) | 0.4754 |
| Grade 3 | 1.58(0.79–3.15) | 0.1941 | 1.67(0.82–3.38) | 0.1570 |
| Unknown | 0.86(0.20–3.76) | 0.8360 | 0.67(0.15–2.97) | 0.5969 |

(*Continued*)

**Table 3.** (Continued)

| Characteristic | Univariate HR | P-values | Adjusted HR | P-values |
|---|---|---|---|---|
| **Endocrine therapy** | | | | |
| AIs alone | 1 | | 1 | |
| Tamoxifen alone | 0.19(0.07–0.51) | **0.0010** | 0.15(0.06–0.41) | **0.0002** |
| **Adherence (PDC* > = 0.8)** | | | | |
| Adherence | 0.25(0.16–0.39) | **< .0001** | 0.24(0.15–0.37) | **< .0001** |
| Non-adherence | 1 | | 1 | |
| **Adjuvant chemotherapy** | | | | |
| No | 1 | | 1 | |
| Yes | 0.75(0.50–1.13) | 0.1722 | 0.80(0.49–1.31) | 0.3789 |
| **Radiation therapy** | | | | |
| No | 1 | | 1 | |
| Yes | 0.66(0.43–1.02) | 0.0606 | 0.72(0.46–1.13) | 0.1519 |

* The proportion of days covered

The landscape of endocrine therapy has experienced a significant evolution, marked by a transition from predominantly using tamoxifen before 2013 to a notable preference for AIs. This shift, accompanied by the rising popularity of AIs, introduces a noteworthy aspect of price discrepancy. Analyzing the medication prices covered by the NHI in 2024 reveals that Tamoxifen (Nolvadex[R]) 10mg, Letrozole (Femara[R]) 2.5mg, Exemestane (Aromasin[R]) 25mg, and Anastrozole (Arimidex[R]) 1mg are priced at NT\$4.8, NT\$24.6, NT\$44.2, and NT\$40.0 respectively.

Given the equivalent effectiveness of the inexpensive tamoxifen compared to the costly AIs, unless specific side effects are a concern, choosing the relatively affordable tamoxifen aligns with economic benefits. Furthermore, with the increasing breast cancer screening rate in Taiwan, many cases with early-stage diagnoses are being identified. However, the rising usage of medications primarily recommended for high-risk patients requiring AIs warrants a thoughtful evaluation of their necessity.

This study observed that patients receiving tamoxifen monotherapy exhibited higher rates of non-adherence than those receiving AIs alone (14.3% vs. 9.3%). These findings align with previous studies by Hsieh et al. [13], Hershman et al. [16], and Cavazza et al. [17]. Poor adherence among tamoxifen users may be attributed to the associated side effects. Tamoxifen is known to cause hot flashes and bleeding problems, which can occur early during therapy and lead to treatment discontinuation or withdrawal [18]. Our study revealed that adherence (PDC > = 0.8) is associated with a 0.29-fold recurrence risk. As observed in Pistilli et al.'s study [19], improved adherence among post-menopausal breast cancer patients can reduce recurrence risk. In addition, the 2016 ACS/ASCO Breast Cancer Survivorship Guideline [20] recommends that primary care clinicians counsel patients to adhere to endocrine therapy. We believe that physicians should take an active role in assessing and encouraging adherence to endocrine therapy. It also emphasizes the collaborative and interdisciplinary nature of modern healthcare practices, underscoring the importance of a team-based approach in ensuring the comprehensive care of patients with breast cancer. In this context, physicians play a crucial role not only in prescribing and monitoring endocrine therapy but also in fostering effective communication and cooperation among various healthcare professionals involved in the patient's care journey.

Many factors can influence whether breast cancer patients experience recurrence. High histological grade and immunohistochemical markers (ER- and PR-negative or HER2-positive) are risk factors for recurrence [21]. This study utilized NHIRD and included adjuvant chemotherapy, radiation therapy and histological grade as covariates to control for interference with recurrence. The HER2-positive breast cancer patients have tumors that grow faster and exhibit a more malignant biological behavior due to the overexpression of the HER2 protein, leading to a higher rate of recurrence compared to other types of breast cancer. Amplification or over-expression of HER2 occurs in approximately 15–30% of breast cancers [22]. The conditions of this study are as follows: Patients newly diagnosed with early-stage breast cancer from 2012 to 2016, aged over 55 at the time of diagnosis, and treated with Tamoxifen or AIs alone for more than one year. Among the population meeting these criteria, there were 1,376 HER2-positive patients with ER-positive status, accounting for 17.8%. To minimize the effect on breast cancer recurrence, this study excludes patients with HER2-positive breast cancer and those who have undergone targeted therapy.

The median follow-up time for endocrine therapy in early breast cancer was 120 months in the ATAC trial [1], 76 months in BIG 1–98 [2], 64 months in ITA [5], and 30 months in MA.17 [23]. In our study, the median follow-up time is 69 months. Most patients (71.6%) were followed up for 4–7 years. Our research team conducted a study [24] using the NHIRD, with enrollment from 2000 to 2005 and follow-up until 2013. In this period, there were only 51 individuals treated with AIs, and 561 individuals treated with tamoxifen. To increase the number of AIs use cases for comparability between the two drugs, enrollment was extended to 2012–2016, resulting in a shorter follow-up period, which is a limitation of this study.

This study utilized the NHI database, which covers nearly the entire population of Taiwan and is highly representative. However, it has the following limitations. Firstly, it can only provide data on insurance-covered treatments, lacking information on patients who obtain medications through out-of-pocket payments. Secondly, due to the higher cost of AIs compared to tamoxifen, the NHI imposes stricter reimbursement criteria for AIs, resulting in patients treated with AIs having a higher disease severity. To mitigate this impact, this study is linked to the TCR database to consider the stage of breast cancer and histological grade. Thirdly, data on the menopausal status of the patients was lacking.

## Conclusion

Based on an analysis of real-world data, the use of tamoxifen in ER+ HER2- post-menopausal women with early breast cancer in Taiwan yields a lower risk of recurrence compared to AIs. This finding suggests that tamoxifen can be an effectiveness treatment option for this population.

Furthermore, it is worth noting that improving medication adherence can disrupt the cycle of breast cancer recurrence. Healthcare providers can optimize the therapeutic benefits and improve recurrence-free survival rates by ensuring patients adhere to their prescribed medication regimen.

## Author Contributions

**Conceptualization:** Chuan-Hsun Chang.

**Data curation:** Yi-Chan Lee, Chun-Wen Huang, You-Min Lu.

**Formal analysis:** Yi-Chan Lee.

**Investigation:** Chuan-Hsun Chang, You-Min Lu.

**Methodology:** Chuan-Hsun Chang, Yi-Chan Lee, Chun-Wen Huang.

**Project administration:** Chun-Wen Huang, You-Min Lu.

**Software:** Yi-Chan Lee.

**Supervision:** Chuan-Hsun Chang.

**Validation:** Yi-Chan Lee, Chun-Wen Huang.

**Writing – original draft:** Yi-Chan Lee, You-Min Lu.

**Writing – review & editing:** Chuan-Hsun Chang, Yi-Chan Lee, Chun-Wen Huang, You-Min Lu.

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
