## [Decision Letter · Decision Letter 0]

26 Dec 2023

PONE-D-23-23146Is tamoxifen good enough for the Asian Population in post-menopausal breast cancer?

A nationwide population-based cohort studyPLOS ONE

Dear Dr. Lu,

Thank you for submitting your manuscript to PLOS ONE. After careful consideration, we feel that it has merit but does not fully meet PLOS ONE’s publication criteria as it currently stands. Therefore, we invite you to submit a revised version of the manuscript that addresses the points raised during the review process.

The topic of comparing the efficacy of tamoxifen vs aromatase inhibitors in the treatment of breast cancer in postmenopausal women is important and interesting. However, the evaluation process of the manuscript brings up important issues that should be taken into account and corrected before tha manuscript is acceptable. Please consider carefully the concise expert review and revise the manuscript accordingly.Especially, as an obtainable patient and tumor information, data on adjuvant chemotherapy and predictive tumor characteristics (ER and HER2 status, proliferation markers, histological grade) should be presented and discussed in the light of tumor responses. Please also discuss a rather short follow-up time as a weakness of the study. Also, use of invasive-free survival instead of overall survival would be advisable as suggested.

We look forward to receiving your revised manuscript.

Kind regards,

Pirkko L. Härkönen, M.D., Ph.D.

Academic Editor

PLOS ONE

Journal Requirements:

"YES

This study was funded by the Cheng Hsin General Hospital (CHGH110-(N)21)."     

"This study was funded by the Cheng Hsin General Hospital (CHGH110-(N)21)."

"YES

This study was funded by the Cheng Hsin General Hospital (CHGH110-(N)21)."

5. We note that you have indicated that there are restrictions to data sharing for this study. PLOS only allows data to be available upon request if there are legal or ethical restrictions on sharing data publicly. For more information on unacceptable data access restrictions, please see http://journals.plos.org/plosone/s/data-availability#loc-unacceptable-data-access-restrictions.  Before we proceed with your manuscript, please address the following prompts:

6. Please amend your manuscript to include your abstract after the title page.

7. Please upload a new copy of Figure 2 as the detail is not clear. Please follow the link for more information: https://blogs.plos.org/plos/2019/06/looking-good-tips-for-creating-your-plos-figures-graphics/’ https://blogs.plos.org/plos/2019/06/looking-good-tips-for-creating-your-plos-figures-graphics/

Reviewers' comments:

Reviewer's Responses to Questions

**Comments to the Author**

1. Is the manuscript technically sound, and do the data support the conclusions?

Reviewer #1: Partly

2. Has the statistical analysis been performed appropriately and rigorously? 

Reviewer #1: Yes

3. Have the authors made all data underlying the findings in their manuscript fully available?

Reviewer #1: No

4. Is the manuscript presented in an intelligible fashion and written in standard English?

Reviewer #1: Yes

5. Review Comments to the Author

Reviewer #1: Dear authors,

Thank you for you manuscript evaluating outcome by type of endocrine therapy. The authors present data from a large cohort evaluating survival in postmenopausal breast cancer patients by type of endocrine therapy.

Minor comments:

The section on age for postmenopausal status can be shortened.

The number of patients at risk should be listed in the Table of outcome by stage.

Major comments:

The follow-up is short to evaluate the true effect of any endocrine therapy, this should be added to the discussion section. The difference in outcome by type of endocrine therapy is hard to evaluate after less than five years of followup.

Moroever, data on adjuvant chemotherapy should be included as well as data on the tumor´s biological inherence. A minimum is data on oestrogen receptor, HER2-receptor and some proliferation estimation such as Ki67 or histological grade. All these factors greatly influence outcome and it was early anticipated that tumor biology could distinguish breast cancer populations who had greater benefit of adjuvant aromatase inhibitors. This variables should also be discussed in the Discussion. A sentence on Taiwan´s nationel guidelines for adjuvant therapy would also be helpful for the reader.

Consider using invasive recurrence free survival or interval instead of overall survival; these endpoints are more relevant as an endpoint in breast cancer trials/retrospective studies.

6. PLOS authors have the option to publish the peer review history of their article (what does this mean?). If published, this will include your full peer review and any attached files.

Reviewer #1: **Yes: **Lisa Rydén, professor Lund University

---

## [Author Response · Author response to Decision Letter 0]

8 Feb 2024

Dear Dr. Härkönen

We have meticulously reviewed and addressed the feedback from the reviewers and the editor. Our responses to each point can be found in the "Response to Reviewers."

Best regards, 

You-Min Lu

---

## [Decision Letter · Decision Letter 1]

22 Feb 2024

PONE-D-23-23146R1Is tamoxifen good enough for the Asian Population in post-menopausal women with early breast cancer?

A nationwide population-based cohort studyPLOS ONE

Dear Dr. Lu,

Thank you for submitting your revised manuscript to PLOS ONE. Although the presentation of the data and the manuscript generally have improved it does not yet, unfortunately, fully meet PLOS ONE’s publication criteria as it currently stands. Specifically, as pointed out in the first review, the status of the breast tumors classified as ER+,HER- should be certified. However, if it is not possible, the reasons and consequences should be thoroughly discussed and the the uncertainly included should be considered as a limitation in drawing conlusions. Secondly, as the external reviewer pointed out, the "targeted therapy" patients should br excluded from the Tam and AI arms because they may critically distort the results. This revision is mandatory. In addition, please consider adding a life-table with the numbers of patients at risk to the Kaplan-Meier plots.If you are able to complete the data as specified above and in the external reviewer's statement, we invite you to submit a re-revised version of the manuscript.

We look forward to receiving your revised manuscript.

Kind regards,

Pirkko L. Härkönen, M.D., Ph.D.

Academic Editor

PLOS ONE

Reviewers' comments:

Reviewer's Responses to Questions

**Comments to the Author**

1. If the authors have adequately addressed your comments raised in a previous round of review and you feel that this manuscript is now acceptable for publication, you may indicate that here to bypass the “Comments to the Author” section, enter your conflict of interest statement in the “Confidential to Editor” section, and submit your "Accept" recommendation.

Reviewer #1: (No Response)

2. Is the manuscript technically sound, and do the data support the conclusions?

Reviewer #1: Partly

3. Has the statistical analysis been performed appropriately and rigorously? 

Reviewer #1: Yes

4. Have the authors made all data underlying the findings in their manuscript fully available?

Reviewer #1: Yes

5. Is the manuscript presented in an intelligible fashion and written in standard English?

Reviewer #1: Yes

6. Review Comments to the Author

Reviewer #1: Thank you for the updated manuscript.

Major comments.

Data on tumor inherence is still lacking, unfortunately. The population should be restricted to those having an ER+/HER2- tumor. By now the variable "targeted therapy" can be used as a surrogate for HER2-status.

Targeted therapy was delivered to 696 of 7116 patients of which 613 (88%) were found in the AI arm. HER2 status is a strong prognostic factor in ER+ disease and the skewed proportion of patients having targeted therapy, implicated as HER2 positive disease, inbetween the two strata of endocrine therapy can putatively explain the improved outcome by tamoxifen. This is real-world data - not a randomized trial - so all baseline data must evaluated.

The limitation of not being able to select patients by ER+/HER2- status should be elaborated on in the Discussion and I do recommend to exclude all patients having "targeted therapy" as a surrogate for HER2 status.

The Kaplan-Meier plots are to be complemented by a life-table showing numbers at risk.

7. PLOS authors have the option to publish the peer review history of their article (what does this mean?). If published, this will include your full peer review and any attached files.

Reviewer #1: No

---

## [Author Response · Author response to Decision Letter 1]

8 Aug 2024

In response to the reviewers' comments, we have taken the following actions to address the concerns raised:

1. Certification of ER+, HER2- Status: We have reapplied for data from the Health and Welfare Data Science Center and obtained detailed information on tumor characteristics. We have now restricted our study population to those with ER-positive and HER2-negative status, as recommended.

2. Exclusion of Targeted Therapy Patients: We have excluded patients who received targeted therapy from our study's Tamoxifen (Tam) and Aromatase Inhibitor (AI) arms to prevent potential distortion of results. This revision has been implemented per the external reviewer's mandatory request.

3. Life-Table Addition: With the suggestion of enhancing our Kaplan-Meier plots, we added a life-table displaying the number of at-risk patients.

---

## [Editor Report · Decision Letter 2]

21 Oct 2024

Is tamoxifen good enough for the Asian Population in ER+ HER2- post-menopausal women with early breast cancer?

A nationwide population-based cohort study

PONE-D-23-23146R2

Dear Dr. You-Min Lu,

We’re pleased to inform you that your manuscript has been judged scientifically suitable for publication The required further analyses have been done properly and discussed and reported appropriately. The manuscript will be formally accepted for publication once it meets all outstanding technical requirements.

The second revision of the manuscript was careful, includes the data and analyses suggested, and appropriate interpretation and and discussion of the results. For my part, I apologize for the unintentional confusion and delay in handling the revised manuscript.

Kind regards,

Pirkko L. Härkönen, M.D., Ph.D.

Academic Editor

PLOS ONE
---

## [Editor Report · Acceptance letter]

18 Nov 2024

PONE-D-23-23146R2 

PLOS ONE

Dear Dr. Lu, 

I'm pleased to inform you that your manuscript has been deemed suitable for publication in PLOS ONE. Congratulations! Your manuscript is now being handed over to our production team.

Kind regards, 

on behalf of

Dr. Pirkko L. Härkönen 

Academic Editor

PLOS ONE